# Density-dependence tips the change of plant–plant interactions under environmental stress

Ruichang Zhang ®[1✉] & Katja Tielbörger[1]

Facilitation studies typically compare plants under differential stress levels with and without neighbors, while the density of neighbors has rarely been addressed. However, recent empirical studies indicate that facilitation may be density-dependent too and peak at intermediate neighbor densities. Here, we propose a conceptual model to incorporate density-dependence into theory about changes of plant–plant interactions under stress. To test our predictions, we combine an individual-based model incorporating both facilitative response and effect, with an experiment using salt stress and *Arabidopsis thaliana*. Theoretical and experimental results are strikingly consistent: (1) the intensity of facilitation peaks at intermediate density, and this peak shifts to higher densities with increasing stress; (2) this shift further modifies the balance between facilitation and competition such that the stress-gradient hypothesis applies only at high densities. Our model suggests that density-dependence must be considered for predicting plant–plant interactions under environmental change.

[1] Plant Ecology Group, University of Tübingen, Auf der Morgenstelle 5, D-72076 Tübingen, Germany. ✉email: ruichang.zhang.ac@gmail.com

Competitive interactions and environmental stress are crucial factors for structuring plant communities and shaping species distributions[1,2]. Though competition dominates the literature, positive plant–plant interactions (facilitation) have been widely explored in the past two decades. For example, facilitation may affect species distribution and range shifts with climate change[3,4] and/or plant diversity from local to large scales[5]. However, most facilitation research is about specific empirical case studies, while theoretical progress helping to generalize the findings has far lagged behind. The most influential model of facilitation, stress-gradient hypothesis (SGH), which predicts dominance of facilitation over competition under intense stress[6], is conceptual instead of quantitative and there is still much debate about its generality[7–15]. Apparently, this hinders our ability to connect plant interactions with environmental change, thus hampering efforts to predict the response of populations and communities to pressures and disturbance under ongoing global change[16–18].

One reason why the theoretical progress has lagged behind could be that most facilitation studies are empirical and only look at target plants under two conditions—with and without neighbors, while the quantity of neighbors (i.e., their density), has been largely ignored. Density dependence has been paramount for theoretical studies of competition and associated fundamental rules such as species coexistence[19–21], the law of constant final yield[22], or self-thinning[23]. It is therefore surprising that facilitation studies rarely address a core aspect of plant interactions i.e., density-dependence[11,24–26].

The density of neighbor plants should have major effects on facilitation[27]. For example, the magnitude of facilitation could increase with benefactor densities, if the net effect of one benefactor is positive. In other words, low densities may be insufficient to alleviate stress in the neighborhood effectively[28,29]. However, stress amelioration is not unlimited (e.g., when neighborhood has been fully shaded by neighbors under hot and dry conditions). Also considering that facilitative and competitive interactions usually co-occur[27,30], then facilitation should be superseded by competition at very high densities. Consequently, we may expect a non-linear relationship between density and net facilitative effects.

Recent empirical studies support this idea and show that benefactors exert the strongest facilitative effect at intermediate densities[28,31–33]. However, few existing studies have attempted to explicitly link density-dependence with changes of plant interactions along stress gradients. Facilitation involves benefactors and beneficiaries, i.e., stress amelioration by benefactors (facilitative effect) and sensitivity of beneficiaries to this amelioration (facilitative response) are equally important[34]. Nevertheless, most studies focus either on beneficiaries or on benefactors alone[35,36] and so are the few studies investigating density-dependence[24,27].

The facilitative response of a beneficiary is determined by the distance to its ecological optimum[17,36,37], i.e., plants are more sensitive to facilitation under intensive stress. Thus, the facilitation–density curve should move upward with increasing stress (Fig. 1a), unless stress exceeds physiological tolerance of beneficiaries. On the other hand, benefactors also experience stress, which may strongly weaken their own performance (e.g., size) and beneficial effects[11,26,38]. We may assume that higher densities compensate the reduced positive effects of smaller (or fewer) beneficiaries, i.e., more neighbors are needed at higher stress levels for maintaining a given level of habitat amelioration. Accordingly, the peak of facilitation will occur at higher densities, i.e., with intensifying stress, the unimodal curve moves to the right along a density gradient (Fig. 1a). This further suggests that at low densities, the net outcome will be less positive and the SGH may not apply (Fig. 1b).

Here, we combine a mathematical model with highly controlled experiments to test these predictions in an intraspecific setting. Classical theoretical studies of negative density-dependence, such as self-thinning[22,23], have been based on a population level. Therefore, any attempt to integrate classical competition theory with facilitation should not avoid the intraspecific setting. Indeed, facilitation may be observed less easily among conspecifics due to larger niche overlap[19,20] and stronger intraspecific competition. However, the advantage is that it could provide a general framework, which is unconfounded by species-specificity of facilitative effect and response[34,39]. Namely, (conspecific) benefactors and beneficiaries do not differ in their niches and are affected similarly by any given stress factor and vice-versa, share the very same resources.

Our individual-based model is designed to provide generic insights into the link between plant interactions and environmental change. The few existing models[31,40] focus narrowly on facilitative effect, while differential response has rarely been considered simultaneously. However, as a pairwise interaction, facilitation should be determined by both of them. Recent empirical studies have also showed that even conspecifics could differ greatly in their response, with smaller ones being more sensitive than larger individuals[41,42]. Moreover, neglecting facilitative response has led to a mathematical problem, where the units on the two sides of their equations are not identical (see "Methods"). By integrating both elements, our model is able to predict the change of plant interactions under stress more explicitly. Though our model is designed to be very general, we also test whether this pattern can be reproduced with real plants. To that end, we conduct a parallel greenhouse experiment with the model plant *Arabidopsis thaliana* which are grown in a full-factorial setting along both a density and a salinity gradient. Salinity is selected as the stress factor, because it is easy to manipulate quantitatively[34] and has been well studied[29,34,43].

We first examine whether facilitation follows the hypothesized hump-shaped relationship. Then we test the following predictions: (1) With increasing stress, facilitation peaks at higher densities. (2) This rightward shift along the density axis changes the balance between facilitation and competition i.e., the SGH holds at high but not at low densities. Results of model simulations and the experiment are strikingly similar and they strongly corroborate these predictions, suggesting the change of plant–plant interactions along stress gradients can be predictable, but only when density is considered explicitly. These findings also indicate the importance of including density-dependence in models for understanding the response of plant populations and communities to environmental change.

## Results

**Density-dependence of plant–plant interactions**. Changes of relative interaction indexes (RIIs) indicated that the net outcome of plant–plant interactions was strongly affected both by stress and density. Note that RII is used to quantify the strength of net plant interactions, which ranges from −1 to 1 with negative values indicating competition and positive values net facilitative interactions (see "Methods"). In model simulations, the relationship between RIIs and density changed from monotonically decreasing to hump-shaped with increasing stress (Fig. 2; see Supplementary Fig. 1 for more stress levels). The experimental results, i.e., from a linear to a hump-shaped relationship, were strikingly similar to those of the model (Fig. 3). Similar patterns were also found for survival and fecundity (Supplementary Figs. 2, 3).

**Shift of the facilitation–density relationship**. With increasing stress, both model and experiment yielded a peak in RII at higher densities. Specifically, the unimodal facilitation–density curve

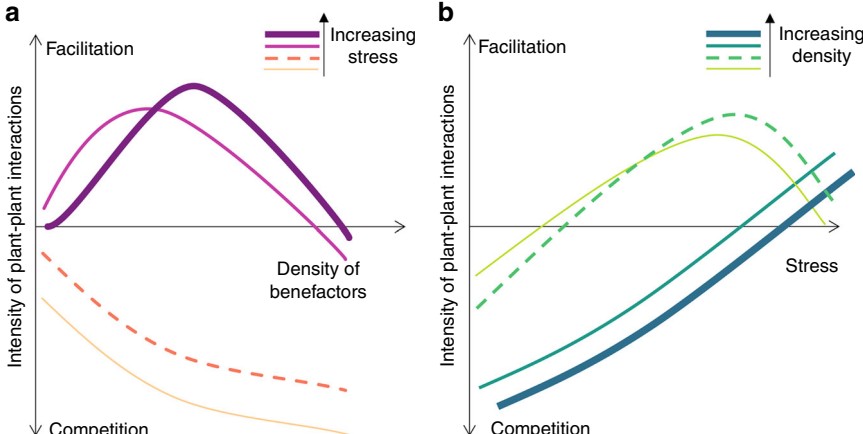

**Fig. 1 The change of plant–plant interactions under stress. a** How interaction intensity–density relationships should shift along a stress gradient and how the shift may affect the balance between facilitation and competition. Under low density (left of the intersection between the two highest stress levels), interactions become less negative with increasing stress but are not most positive for the highest stress level, whereas under high density, the shift in interaction intensity is continuous and follows the predictions of the stress-gradient hypothesis. **b** Same model as panel **a** but displaying how plant–plant interactions at differential densities change along a stress gradient. This shows that the SGH applies only to relatively high densities.

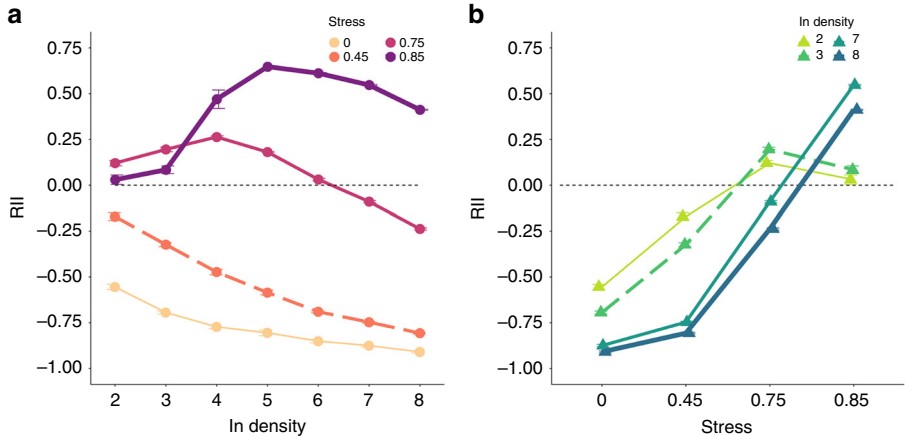

**Fig. 2 The change of density-dependent interactions under stress in model simulations. a** Relationship between initial density and relative interaction index (RII; data are presented as mean values ± SEM) in simulated populations growing along a stress gradient. **b** Plant–plant interactions change along the stress gradient, showing that the SGH applies at high densities but not at low densities. Circles represent different stress levels while triangles represent different density levels. $N = 5$ independent samples in model simulations for the density gradient from two to eight at each stress level. Note that for facilitating visual interpretation, we only show the two lowest and two highest densities (changes in the RII for intermediate densities were similar and ranged in between the extreme scenarios shown here). Source data are provided as a Source Data file.

shifted to the high end of the density gradient (Figs. 2a, 3a). Correspondingly, at high densities (e.g., 15 and 20 individuals per pot), net plant–plant interactions shifted from more negative to more positive along the stress gradient (Figs. 2b, 3b). However, at low densities (e.g., two and three individuals per pot), facilitation under high stress was more intense than under extreme stress, i.e., the balance between positive and negative interactions responded in a hump-shape to increasing stress levels (Figs. 2b, 3b).

The Bayesian analysis further showed that the full model including density and stress (and their interactions) provided the best-fit to the dataset for both simulations and the experiment (Table 1). There was an approximated probability of 95% for the full model accurately predicting the change of RIIs–density relationship along stress gradients, and this also indicated that the RIIs–density relationship at different stress levels indeed differed greatly (Fig. 4). Furthermore, estimates based on model-averaging were generally similar to the full model. However, its credible sets were larger, reflecting greater uncertainty (Supplementary Fig. 4).

The estimated model parameters also fall within the 95% credible sets and they were shown in the Supplementary information (Supplementary Table 1).

**Effects of spatial patterns, modes, and growth.** In additional simulations, the intensity of facilitation at relatively high stress levels was generally stronger with an aggregated spatial pattern than that under regular spacing, but neither spatial pattern nor different modes of competition and facilitation affected the facilitation–density relationship qualitatively (Supplementary Fig. 5). Similarly, the same general pattern was observed when considering the growth of plants, only that the facilitation–density curve shifted leftward along the density gradient over time, i.e., RIIs peaked at lower densities (Supplementary Fig. 6a, b). RIIs generally increased with size (biomass) of plants during growth, except for individuals at relatively high densities. These plants exhibited a unimodal facilitation–size

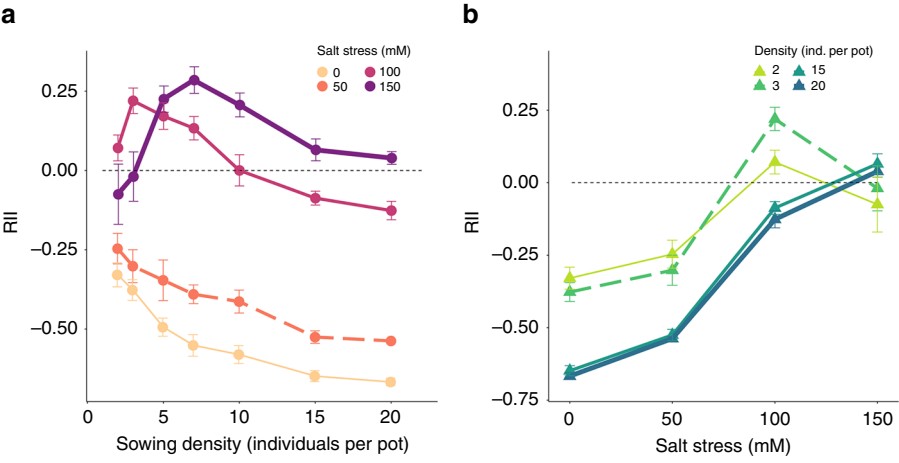

**Fig. 3 The change of density-dependent interactions under salt stress in the experiment. a** Relationship between initial density and average relative interaction index (RII; data are presented as mean values ± SEM) for biomass of *Arabidopsis thaliana* grown along a salinity gradient in a greenhouse experiment. **b** Plant–plant interactions change along the stress gradient, showing that the SGH applies at high densities but not at low densities. Circles represent different stress levels while triangles and lines represent different density levels. For the density gradient from 2 to 20 plants per pot, $n_{salt\_0}$ = 25, 29, 17, 16, 15, 15, and 16 independent pots, respectively; $n_{salt\_50}$ = 20, 25, 13, 17, 16, 13, 16; $n_{salt\_100}$ = 30, 26, 15, 12, 14, 14, 13; $n_{salt\_150}$ = 29, 30, 15, 16, 17, 12, 11. Note that for facilitating visual interpretation, we only show the two lowest and two highest densities (changes in the RII for intermediate densities were similar and ranged in between the extreme scenarios shown here). Source data are provided as a Source Data file.

**Table 1 Bayesian models relating plant–plant interactions to stress and density.**

| Models | WAIC (simulations) | WAIC (experiment) |
|---|---|---|
| RIIs = S | −59.6 | 37.4 |
| RIIs = D | 218.4 | 386.2 |
| RIIs = S + D | −80.5 | −0.1 |
| RIIs = S + D + S × D | −524.4 | −53.8 |

WAIC is used to compare these Bayesian models in model simulations and the experiment, respectively, and the model with smaller WAIC value is considered to be a better model. *RII* relative interaction index, *S* stress in model simulations or salinity level in the experiment, *D* density, *WAIC* Watanabe–Akaike information criterion.

relationship, i.e., facilitation first increased with size while competition dominated as plants grew bigger (Supplementary Fig. 6c, d).

## Discussion
Overall findings provide unequivocal support for our initial predictions. There was a predictable, density-dependent relationship between interaction intensity and direction and stress levels, and a predictable shift towards higher densities, which was either consistent or against SGH[6]. All results were strikingly consistent between the experiment and the model (which incorporated a wide range of scenarios), indicating that they are highly robust.

It is not surprising that monotonous negative density-dependence dominated under relatively benign conditions. This pattern has long been considered universal and is a cornerstone of population biology[44]. Despite the overwhelming evidence for the ubiquity of positive interactions[45], facilitation has been ignored in most models addressing density-dependence. Only few models have considered density-dependence of facilitation[31], and despite some limitations (see "Methods") these corroborate one of our findings, namely that the maximum benefit from neighbors was obtained at intermediate densities. The generally weak positive interactions under low density can be explained by the fact that plants are too remote from each other to affect their neighbors.

Though our findings challenge theories based solely on competition[22,46], they are consistent with recent empirical studies in stressful habitats. For instance, strong support for a unimodal fecundity–density relationship was found in a Tibetan lotus species[32]. There was also evidence of density-dependence from interspecific cases, e.g., Dickie et al. found that with increasing density of *Quercus ellipsoidalis*, the growth of seedlings (*Quercus macrocarpa*) first increased and then declined[33].

Although the hump-shaped response of facilitation to density has been suggested by a few previous studies, confirming this relationship was important because it served as the assumption underlying our main prediction of an upward and rightward shift of the non-linear facilitation–density curve under prevailing stress.

Indeed, RIIs peaked at larger densities with increasing stress (i.e., rightward shift) in both simulations and the experiment. This may be explained by the fact that at higher stress levels, more neighbors are required for ameliorating habitat conditions due to two non-mutually exclusive mechanisms. On the one hand, if a benefactor of a given size alleviates stress to a certain level, then more plants are needed to generate the same conditions in more severe environments. On the other hand, positive effects of benefactors may decline because they themselves are stressed and smaller[11,26,38]. This could also happen dynamically in time during plant growth. Model simulations showed that facilitation generally increased with plant size during growth, unless there were too many neighbors. Intriguingly, we further found that the facilitation–density curve moved towards the low end of the density gradient over time (i.e., RIIs peaked at lower densities), indicating that fewer plants are needed for ameliorating stressful conditions as plant size and their positive effects increased. Therefore, the dynamics of facilitation also corroborated our hypothesis that the curve should shift rightward when facilitative effects are decreased by stress, abelite indirectly.

If we apply these ideas to our experiments, we must consider that mechanism by which plants facilitate each other under salt stress can be either reduced evaporation via shading of the substrate or salt uptake from the soil[29,34]. Clearly, this facilitative effect should become stronger with more and/or larger neighbors present, until the vicinity of beneficiaries is fully shaded by

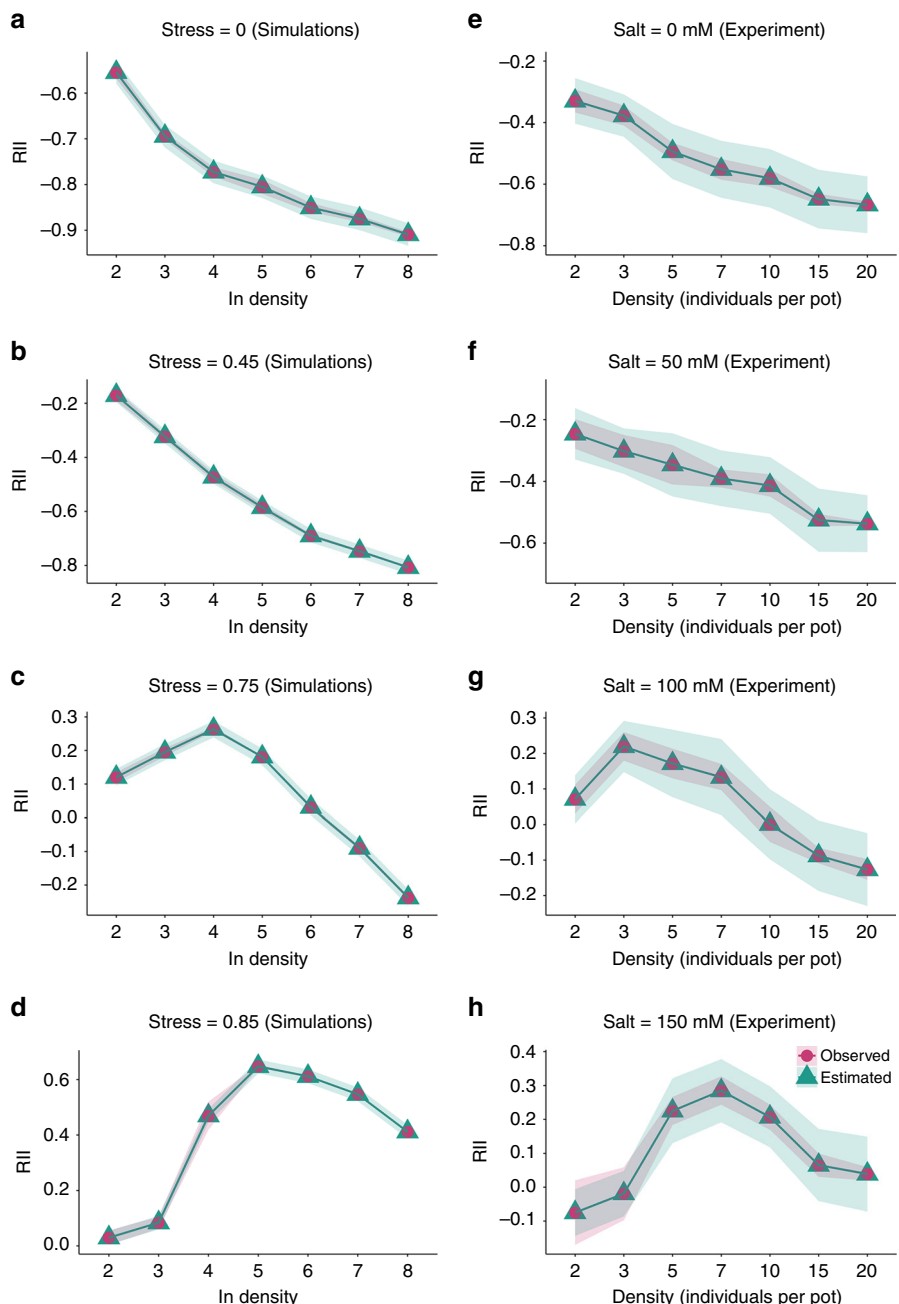

**Fig. 4 Bayesian estimates and observed plant–plant interactions. a–d** Bayesian estimates (with 95% credible set) and observed average index (RII; with 95% confidence interval) along a density gradient at each stress level in the simulations ($n = 5$ independent samples in model simulations for the density gradient from two to eight at each stress level). **e–h** experimental data (for the density gradient from 2 to 20 plants per pot, $n_{salt\_0} = 25, 29, 17, 16, 15, 15$ and 16 independent pots, respectively; $n_{salt\_50} = 20, 25, 13, 17, 16, 13, 16$; $n_{salt\_100} = 30, 26, 15, 12, 14, 14, 13$; $n_{salt\_150} = 29, 30, 15, 16, 17, 12, 11$). Circles and represent observed values while triangles and lines represent Bayesian estimates. Source data are provided as a Source Data file.

canopy or the salt is fully retrieved by neighbors. In our study, performance of all individuals was greatly limited by salinity, and thus more neighbors were needed for alleviating salt stress. Though we tested our model with a single species and stress type (salinity), our simulations provided also the theoretical explanation for studies addressing many other stress factors. It is reasonable to assume that if benefactors ameliorate different stressful conditions, e.g., drought (via shading), disturbance (via substrate stabilization), herbivory (via physical or chemical defense), soil contaminants (via uptake), and many more, then more and/or larger neighbors would yield a larger facilitative effect, too. There are also a few empirical studies exemplifying that our

experimental findings are unlikely to be exclusive for salt stress. For example, Goldenheim et al. found for a gradient of evaporative stress that the forb *Suaeda linearis* switched from negative to positive density-dependence, i.e., in more stressful conditions, plants showed greater biomass and seed production at higher densities due to mitigated desiccation and thermal stress by neighbors[47]. Similarly, Bos and van Katwijk reported that as hydrodynamic exposure increased, survival of eelgrass *Zostera marina* was significantly higher in the high-density group because neighbors could reduce drag force when exposed to currents[48].

Our second main prediction was that due to the above shift in the unimodal relationship, the balance between competition and

facilitation is density-dependent. There was again strikingly consistent support for this hypothesis from the experiment and modeling. Specifically, due to the shift of the facilitation–density curve, the SGH[6] was supported for high densities, where interactions changed from predominantly negative to positive with increasing stress. Despite the fact that plant performance was decreased under high densities and intense stress, the number of benefactors was still sufficient for ameliorating the stress, i.e., even the area shaded by each plant was reduced by salt stress in the experiment. However, this pattern was not confirmed under low density and high stress. Under such conditions, not only the facilitative effect of each individual benefactor but also their number was too small. Therefore, initially positive interactions could shift towards neutral or negative.

In fact, many empirical findings have reported the decreased facilitation along stress gradients[9–15,26,38]. Nevertheless, only case-specific explanations have been proposed and the link to density-dependence has not been made[12,14]. Indeed, most previous studies merely compared the performance of target plants under two density levels only (with and without neighbors), while models incorporating stress gradients and density-dependence of competition and facilitation are virtually missing. Therefore, our findings may help to provide a quantitative framework for predicting under which conditions the SGH should apply. Such a quantitative framework is urgently needed because there is still no consensus about the generality of the SGH[8,25,26,35]. This also prevents us from predicting how plant populations and communities may be affected by environmental change[17]. Here, we demonstrate that by explicitly considering the density of neighbors, it is possible to predict the outcome of plant–plant interactions under different levels of stress.

Our model adopts the idea that facilitation is determined by both facilitative effect and response. However, to our knowledge previous models only considered facilitative effect[31,40,49–51]. This is regrettable because even for the same species, smaller and/or younger plants respond more positively to stress mitigation than larger ones[42]. In addition, much accepted theory related to sensitivity to facilitation is also based on species-specific stress tolerance[37] which in turn has been related to size in classical plant strategy theory (Grime's CSR). It is therefore surprising that facilitative response has not been considered in the few existing models. This has also resulted in a mathematical problem, i.e., unbalanced units on the two sides of equations[31,40]. In fact, these models yield findings which not only contradict our experiments but also cannot be explained. Namely, the facilitation–density relationship moves upward or leftward with increasing stress and one model even indicates strong competition at very low densities under high stress[31]. This is counterintuitive, and cannot be explained by decreased facilitation intensity with increasing stress at low densities.

It should be noted that while we selected an intraspecific setting for providing a general framework, there could obviously be species-specific differences between benefactors and beneficiaries in interspecific settings, e.g., benefactors and beneficiaries may differ dramatically in stress tolerance, shape and other functional traits[17,25]. Our model is based on integrating both effect and response because multiple drivers of facilitation could be encapsulated into the two general factors. Therefore, those differences may also determine the shift of facilitation–density relationship, if they could affect the decline of facilitative effects (or response) along stress gradients. For example, benefactor species may be highly tolerant to or even favor, up to a certain point, the very stress factor limiting the beneficiaries[39]. In this scenario, if net effects of benefactors were not reduced by stress, we would expect that the curve simply moves upward (or perhaps leftward with increased facilitative effects) instead of moving to the right.

Furthermore, an interspecific scenario would also differ in the way negative plant–plant interactions play out, i.e., competition should be less intense between than within species[19]. In this case, the RII–density relationship would again be similar to our intraspecific case, but facilitation may dominate across a larger range of densities and the peak of the curve should be generally higher. We thus suggest that our model may yield the basis for further exploring of density-dependence between different species.

Our overall findings indicate that plant–plant interactions change predictably—but partially differently than suggested by the SGH—along stress gradients. The striking consistency between our quantitative model and experiments indicates that the findings are robust. Including density-dependence in models of facilitation will undoubtedly promote to predict more precisely how vegetation could respond to environmental change. Furthermore, the findings could provide insights into species coexistence hypotheses, which mainly focus on competition and often assume that the overall effect of one species on another is linearly and negatively related to its density[19,20]. However, we show that unimodal relationships should occur and that these depend on stress. Our findings could also be helpful for ecological restoration and crop production[29]. Because of the universal nonoptimal habitat conditions and ongoing environmental change[17,18], we advocate the inclusion of density as a factor mediating positive and negative plant interactions in experiments and models. Our study represents a robust contribution to such an approach and may help to reconcile competition and facilitation theory into a common framework.

## Methods
**Model**. A detailed model description is provided in the Supplementary methods. Here, we only illustrate the core equations. In the model, an individual's potential growth rate (without neighbors) is defined as:

$$\frac{dm}{dt} = am^{\frac{3}{4}}\left[1 - S - \left(\frac{m}{M}\right)^{\frac{1}{4}}\right] \tag{1}$$

where $m$ is total mass, $a$ is a species-specific constant, and $M$ is the theoretically maximum mass. $S$ is the stress intensity quantified as the proportional reduction in incoming energy. It ranges from 0 (no stress) to 1 (extreme stress and no resources).

The equation is based on Metabolic Scaling Theory[52] and it was integrated with the zone-of-influence (ZOI) model to include interactions[53]. In the spatially-explicit ZOI model, each plant is modeled as a circle and its size ($A$, unit: area) denotes potentially available resources and is allometrically related to the biomass, as $A = C_0 m^{3/4}$, where $C_0$ is a constant (see Supplementary methods). Individuals interact with each other where their ZOIs overlap. Therefore, the realized growth rate is as follows:

$$\frac{dm}{dt} = rA\left[(1 - SI_f)I_c - \left(\frac{m}{M}\right)^{1/4}\right] \tag{2}$$

where $r$ indicates the intrinsic growth rate (in mass area$^{-1}$ time$^{-1}$) while $I_c$ and $I_f$ are competition and facilitation indexes, respectively. $I_c$ represents the proportion of actual resources available for an individual under competition:

$$I_c = \frac{A_c}{A} \tag{3}$$

where $A_c$ represents the actual resources a plant can obtain. It is calculated as the sum of the nonoverlapping area and the effective that can be obtained within overlapping zones. The division of overlapping areas is determined by the parameter $p$, i.e., the mode of competition[53]. It ranges from 0 (complete symmetry) to $\infty$ (complete asymmetry). In harsh conditions, the size of ZOI also reflects the quantity of perceived stress and stress mitigation occurs in overlapping areas. Similarly, $I_f$ is the proportion of realized stress for an individual experiencing facilitation:

$$I_f = 1 - \frac{A_f}{A} \tag{4}$$

where $A_f$ is a quantitative measure for stress reduction by neighbors. Likewise, the calculation is based on how individuals share overlapping ZOIs (stress), which is determined by the parameter $q$, i.e., the mode of facilitation[51]. It also ranges from 0 (complete symmetric facilitation) to $\infty$ (complete asymmetric facilitation). See also Supplementary methods for detailed calculation process about $I_c$ and $I_f$.

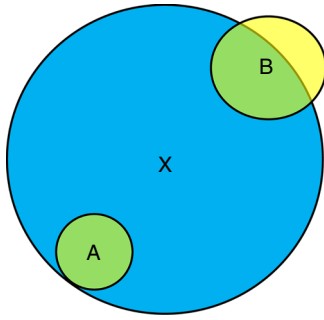

**Fig. 5 Schematic illustration of stress amelioration for plants with different size in a zone-of-influence (ZOI) model. A** and **B** (yellow ZOIs) are two conspecific beneficiary plants and they are facilitated by a benefactor X (blue ZOI). The stress intensity (e.g., salinity) per unit area is homogeneous across space. The area of overlap of ZOIs (green) is the same for both beneficiary plants ($A_{fa} = A_{fb}$). However, plant A fully benefits from the presence of X while B benefits only partly. Thus, beneficiary A will respond more positively to habitat amelioration than beneficiary B, i.e., the facilitative response is size dependent.

Compared to previous models, a main improvement is the explicit consideration of the impact of the size of the beneficiary on its facilitative response. Previous studies have used the term $S/(1 + A_f)$ for effects of stress and facilitation on plant performance[31,40,49–51]. For a given stress level, this term is determined only by $A_f$ i.e., positive effects of benefactors. However, this term is not realistic because it cannot reflect differential sensitivity of beneficiaries with different size to stress amelioration (facilitative response, see Fig. 5 for example). Thus, the influence of facilitation was identical: $S/(1 + A_{fa}) = S/(1 + A_{fb})$, as long as stress alleviation by benefactors was the same ($A_{fa} = A_{fb}$), although the smaller plant should experience stronger facilitation. A fundamental problem arising from that approach is that the unit of $A_f$ (area$^{-1}$) is not balanced in the term $S/(1 + A_f)$, and units on both sides of the growth equation are not identical. Namely, the units are mass time$^{-1}$ on the left side but mass time$^{-1}$ area$^{-1}$[40,51] or even more complicated on the right side[31,50]. Here, we included facilitative response via the term $SI_f = S(1 − A_f/A)$, which is both more realistic and mathematically correct (equal units: mass time$^{-1}$).

The simulations were done in Netlogo (version 6.0.1)[54] using a grid of homogeneous patches ($200 \times 200$) in a continuous two-dimensional space[53] with a wrap-around approach to avoid edge effects[55]. Data were collected every ten time steps and results presented after 50 steps. Plants experienced a density gradient with seven levels from 2 to 8 (in scale), which also encompassed the density range of our experiment, Namely, in the control (without stress), the competition intensity at the minimum and maximum density level in simulations should be lower and higher than that in the experiment, respectively. Following previous studies[22,31], density refers to initial density and not surviving density. We explored a wide range of stress levels and for ease of visualization, present here four of them: no (0), low (0.45), high (0.75), and extreme (0.85), respectively. Other stress levels yielded the same qualitative results (Supplementary Fig. 1). We also evaluated potential effects of spatial distribution (random, regular, aggregated) and different modes of competition and facilitation, but these did not change the qualitative findings either (Supplementary Fig. 5). In the following, we thus mainly present results for the scenario most similar to the intraspecific experiment, i.e., random distribution, symmetric competition ($p = 1$) and symmetric facilitation ($q = 1$). Each simulation was repeated five times. Considering that plant–plant interactions may change over time due to individual growth, we also ran additional simulations. Thus, in each time step, we collected data of the mean biomass under relatively high stress where strong facilitation occurred, i.e., $S = 0.75$ and $0.85$.

The RII was used to quantify strength and direction of net plant–plant interactions[56]:

$$RII = \frac{P_w - P_s}{P_w + P_s} \qquad (5)$$

where $P_W$ and $P_S$ are the performance of plants with and without neighbors, respectively. RII ranges from −1 to 1 with negative values indicating competition and positive values net facilitative interactions. To estimate biomass of individuals without neighbors ($P_S$), we calculated the growth rate as follows:

$$\frac{dm}{dt} = rA\left[1 - S - \left(\frac{m}{M}\right)^{1/4}\right]. \qquad (6)$$

**Experiment**. The experiment was set up in a greenhouse at Tübingen University between early October and mid December 2015 (light intensities: 130–150 µmol m$^{-2}$ s$^{-1}$ and 16 h day$^{-1}$; temperatures: 15 °C (night) and 20 °C (day)). We selected the

model plant *Arabidopsis thaliana* as study species because it is annual and fast-growing, densities can be easily manipulated, and standardized seed material enables to conduct experiments with essentially similar plants. Moreover, previous research has revealed strong intraspecific facilitation under salt stress for this species[34] such that the general system was well established. Seeds of *A. thaliana* (popular *Columbia* wildtypes) were obtained from Nottingham Arabidopsis Stock Centre and inbred lines raised for one generation to amplify seed numbers.

Seeds were sterilized in 70% ethanol for 2 min and stratified at 4 °C for 5 days[57]. They were then transferred to pots ($10 \times 10 \times 10$ cm) with standard potting soil (Einheitserde Classic, Gebr. Patzer Company). Seeds were sown to obtain a density gradient, i.e., 1 (without interactions), 2, 3, 5, 7, 10, 15, 20 seedlings per pot, respectively. Considering the small pots, 20 individuals per pot was actually very dense (i.e., 2000 plants per square meter). Approximately twice the number was sown in each pot for making sure that the desired seedling numbers can be obtained even under incomplete germination. We further prepared 40 additional pots with 10 seeds each as backup. Three days after germination, randomly selected surplus plants were removed. 60 spare individuals were transplanted to pots with insufficient germination. Plants growing with fewer neighbors may suffer more from stress[58], and we observed high mortality of single plants under high salinity in our pilot study. We therefore set more replicates for low densities to obtain enough individuals for the analyses, i.e., 50 for density 1, 30 for 2 and 3, and 20 for each of the higher densities, respectively.

We applied the following four treatments: 0 (no salt), 50 (low), 100 (high), and 150 mM NaCl (extreme) solution. They were selected based on a previous study[34] and they were meant to capture the entire possible range from no stress to extreme stress i.e., 80% limitation in performance such as growth and/or survival for plants without neighbors[35]. Naturally observed stress levels in saline habitats are well within this range, e.g., highly saline conditions may limit growth by up to about 70–75%[29,43]. Salt was first applied eight days after germination and plants were watered every 5–7 days depending on soil water conditions. Each pot received the same amount of liquid at each irrigation event, and the amount was successively increased from 50 ml to 150 ml as plants grew. The location of pots assigned to different treatments was fully randomized. At the end of the experiment, we counted the number of survivors in each pot and determined the number of siliques and above-ground biomass (dried at 70 °C for 24 h) per survivor.

RIIs were also calculated according to Eq. (5). Since the experiment was an intraspecific setting, we calculated $P_W$ and $P_S$ as the mean individual performance per pot without and with interactions, respectively[31].

**Statistical analysis**. We applied Bayesian inference and fitted different models to the dataset of RIIs of simulations and the experiment, respectively (sample size in each treatment were provided in Source data). To compare our own conceptual model with existing theory, four models (Table 1) were fitted: (1) only stress as explanatory variable, which corresponds to classical facilitation models, i.e., SGH, a unidirectional change of RII along a stress gradient; (2) only density, i.e., classical competition models; (3) both density and stress (without the interaction term), i.e., the model that the RII–density relationship simply moves upward with increasing stress); (4) density, stress and their interactions (our own model).

When calculating the joint posterior distribution and uncertainty intervals of each model parameter, we selected uninformative Gaussian priors because there were few previous studies considering how density-dependence of facilitation could change along stress gradients[59]. In the calculation, Markov chain Monte Carlo (MCMC) were applied with 20,000 iterations[60]. After the estimation of posterior, we further investigated the convergence, i.e., whether MCMC chains converged to the posterior ($\hat{R} < 1.05$). The Watanabe–Akaike information criterion (WAIC) was used for comparing these Bayesian models and the model with the smallest WAIC value was selected to be the best model[61]. The uncertainty in model selection was also evaluated, by using weighted averages of all the models[59,62]. All Bayesian statistics and data calculations were performed in R (version 3.6.1)[63] using the package brms[60], which is based on the probabilistic programming language Stan[64].

**Reporting summary**. Further information on research design is available in the Nature Research Reporting Summary linked to this article.

## Data availability

The source data underlying Figs. 2a, b, 3a, b, and 4a–h, Supplementary Figs. 1, 2a–d, 3a–d, 4a–h, 5a, b, and 6a–d, Table 1 and Supplementary Table 1 are provided as a Source Data file. The file is available at GitHub: https://github.com/Halili-z/density-dependence.

## Code availability

Codes are deposited into a public repository (GitHub): https://github.com/Halili-z/density-dependence.

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

## Acknowledgements

We are grateful to Dr. Xiangzi Zheng for his help in the greenhouse. We also thank Dr. Yue Lin for sharing his codes. R.Z. was supported by the China Scholarship Council (CSC).

## Author contributions

R.Z. and K.T. conceived the ideas and wrote the manuscript; R.Z. performed the modeling work and experiments and analyzed the data.

## Competing interests

The authors declare no competing interests.
