## [Peer Review File · Nature Communications]

Reviewers' comments:

Reviewer #1 (Remarks to the Author):

In this manuscript, the authors proposed and tested, in both experiments and simulations, a conceptual model that incorporates density dependence into the stress gradient hypothesis on changes in positive and negative plant interactions with stress. Density dependence has been well established in population biology and theory on competition. Although facilitation has been largely recognized with a rapidly growing body of literature over the last few decades, whether facilitative plant interactions are density-dependent and whether density dependence can help predict how plant-plant interactions change with stress is understudied. In this regard, the work reported in this manuscript made an important step forward and could be a novel contribution to this field.

Although I am not competent to evaluate the modelling component of this work, their experimental component was generally well conducted, the results appear to be convincing, and the manuscript is generally well written. My major and minor concerns are provided below.

Major

1. The authors' models and experiments deal with ONLY intraspecific plant-plant interactions. This must be made very clear in the manuscript throughout. Although the authors argued that their conceptual model and findings should be generally applicable to interspecific settings, this lacks empirical and theoretical support. Actually, interspecific facilitation involves many more factors (e.g., functional traits, plant size, and life stage) that often fundamentally differ between benefactors and beneficiaries. Such factors can importantly determine the nature and intensity of plant-plant interactions and how they change over stress gradients and are not incorporated in their simulations or experiment. So extrapolating to interspecific interactions must be made more carefully.

Many of the statements and discussions in the manuscript mixed studies/citations on intraspecific and interspecific interactions. The manuscript can be greatly improved by focusing on intraspecific interactions only and then discussing about potential generality for interspecific interactions in one or two specific paragraphs.

2. The argument that previous facilitation studies have treated neighbor effects qualitatively and not quantitatively is unfounded. The intensity of neighbor effects has been well quantified in numerous studies. Maybe the authors meant that facilitation studies haven't widely considered density dependence, but not considering density dependence does not mean that neighbor effects haven't been treated quantitatively. Since this is an argument that the authors repeatedly addressed in their manuscript, it needs to be reconsidered.

Minor

Ln 62, there are no explicit models?

Ln 80-81, unclear, and needs to be rewritten

Figure 1. I think another panel showing how the intensity of plant interactions at different plant densities changes along stress gradients would be very helpful.

Ln 97, delete "with"

Ln 102, has been neglected?

Ln 118-122, I think extrapolating to interspecific settings simply based on that intraspecific competition is often stronger than intraspecific competition is more than a stretch.

Figure 2, why was e^8 the maximum density used in simulations, not the exact same range of density as in the experiment? How were the ranges of density comparable between simulations and experiments? There is a short explanation (Ln 344-345), but not sufficient.

Figure 3, Panel B shows how RII changes with salt stress at some of the lowest and highest densities. But how about intermediate densities? Why was it meaningful to have both densities 2 and 3 in panel B, but not density 7 or 10? If this was to match the densities given in Figure 2, it needs to be explained why the range of plant density differed between the simulations and the experiment

Ln 163, Perhaps, the "intermediate" stress here is actually a "high" stress, while the "high" stress here is an "extreme" stress. Note what it really means an intermediate, high, and extreme stress.

Ln 169-175, Why the authors conducted these analyses and presented these results are not aforementioned, so unclear why it's necessary to have this paragraph here.

Ln 194, this is a study on interspecific facilitation.

Ln 225, this is a very bold statement.

Reviewer #2 (Remarks to the Author):

I find the topic interesting, but I have two major concerns with the approach used in the manuscript.

As is briefly mentioned in lines 103 to 109, the plant-plant interactions are not static through time. However, density is a static variable, and will probably only work as a descriptive variable for annual plants. Consequently, it would have been an advantage to consider plant growth data instead of number of seed as the dependent variable. I would also like to see a thorough discussion of this shortcoming in the data and model development.

It would have been a clear advantage to fit the data to the model instead of just comparing the results using graphical methods. In a Bayesian framework, the joint posterior distribution of parameters and uncertainties could have provided important insights into the studied mechanisms.

Reviewer #3 (Remarks to the Author):

The authors combined an individual based model and a greenhouse experiment to explore how neighbour densities may alter the outcome of plant-plant interactions along an environmental stress gradient.

The authors conclude that 1) the maximum benefit from neighbouring plants (i.e. facilitation) peaks at intermediate plant density; 2) this peak shifts towards higher densities if environmental stress increases. The authors find these results to be robust and universally applicable and propose a conceptual model to generalize these findings.

These claims are strongly overstated. The authors used a very particular experimental set up with individuals of one plant species exposed to increasing levels of salinity and neighbour densities. It is quite obvious that plants can benefit from increasing the number of neighbouring plants when exposed to increasing salinity gradients because salt uptake by neighbours reduces salt concentrations in the soil and therefore ameliorates this abiotic stress. The results from this experiment are very specific to salt and cannot be generalized to other type of abiotic (e.g. drought, nutrient limitation) or biotic stresses (e.g. herbivory).

The authors reason that if a benefactor of a given size alleviates stress to a certain level, then more plants are needed to generate the same conditions in a more severe environment. This explanation is quite specific to stresses such as salt and perhaps some pollutants but not for other abiotic stresses. For example, if the authors would have exposed plants of the same size to

increasing neighbour densities along a soil water gradient they would have found increasing competitive effects as soil water decreases.

Point-by-point response to reviews (reviews in *italics*, response indicated by 'R'):

Reviewer 1:

In this manuscript, the authors proposed and tested, in both experiments and simulations, a conceptual model that incorporates density dependence into the stress gradient hypothesis on changes in positive and negative plant interactions with stress. Density dependence has been well established in population biology and theory on competition. Although facilitation has been largely recognized with a rapidly growing body of literature over the last few decades, whether facilitative plant interactions are density-dependent and whether density dependence can help predict how plant–plant interactions change with stress is understudied. In this regard, the work reported in this manuscript made an important step forward and could be a novel contribution to this field.

Although I am not competent to evaluate the modelling component of this work, their experimental component was generally well conducted, the results appear to be convincing, and the manuscript is generally well written. My major and minor concerns are provided below.

R: We are grateful for this overall positive assessment of our work.

Major

1. The authors' models and experiments deal with ONLY intraspecific plant–plant interactions. This must be made very clear in the manuscript throughout. Although the authors argued that their conceptual model and findings should be generally applicable to interspecific settings, this lacks empirical and theoretical support. Actually, interspecific facilitation involves many more factors (e.g., functional traits, plant size, and life stage) that often fundamentally differ between benefactors and beneficiaries. Such factors can importantly determine the nature and intensity of plant–plant interactions and how they change over stress gradients and are not incorporated in their simulations or experiment. So extrapolating to interspecific interactions must be made more carefully.

Many of the statements and discussions in the manuscript mixed studies/citations on intraspecific and interspecific interactions. The manuscript can be greatly improved by focusing on intraspecific interactions only and then discussing about potential generality for interspecific interactions in one or two specific paragraphs.

R: We are grateful for this comment and fully agree with it. As suggested, we now more clearly start out with a focus on intraspecific interactions and explain in more detail in the introduction why we are doing this (**lines 110-119**). We have also, throughout the manuscript, made a much more careful distinction between intraspecific and interspecific interactions and we hope it is much clearer now (examples: **lines 109-110, 260-261, 295-296, 359-360**).

It is true benefactors and beneficiaries may differ dramatically in many species-specific traits. Considering that facilitation is determined by both effect and

response, these differences may influence how facilitative effects (response) change with stress and further determine the shift of facilitation–density curve. We fully agree with this but only simply mentioned it in the original manuscript that the pattern could be different “if net effects of benefactors were not reduced by stress”. Therefore, in the revised version, we have now expanded the discussion into that direction and elaborate cases about how differences between benefactors and beneficiaries might modify the shape of the facilitation–density curves. Additionally, we discuss how reduced competitive interactions, as should be pertinent to an interspecific setting, would change the location of the peak of the RII–density curve but not the general shape **(lines 359-376)**. Multiple drivers that shape facilitation can be encapsulated into the two general factors i.e., effect and response. Then our model essentially shows that irrespectively of species identity, stress reduces facilitative effects and the facilitation–density curve could shift rightward. Thus, by discussing the applicability of our model to interspecific interactions, we hope that the model serves as a basic framework onto which future work can build.

2. The argument that previous facilitation studies have treated neighbor effects qualitatively and not quantitatively is unfounded. The intensity of neighbor effects has been well quantified in numerous studies. Maybe the authors meant that facilitation studies haven't widely considered density dependence, but not considering density dependence does not mean that neighbor effects haven't been treated quantitatively. Since this is an argument that the authors repeatedly addressed in their manuscript, it needs to be reconsidered.

R: We are grateful for this comment because it made us realize that our wording was awkward and misleading. It is true that we meant to say that previous studies only look at the “presence” instead of the “quantity” of neighbors (i.e., density–dependence). We have now spelled this out explicitly and tried to avoid using the notion ‘qualitative vs. quantitative’. To that end, we have also rewritten the abstract and thoroughly revised other sections regarding that point. (examples:**lines 8-9, 21-22, 43-44, 157-158, 336-337,380-381**)

Minor

Ln 62, there are no explicit models?

R: The sentence has been rewritten. **(lines 69-70)**

Ln 80–81, unclear, and needs to be rewritten

R: Done as requested, we hope it is clearer now. **(lines 88-89)**

Figure 1. I think another panel showing how the intensity of plant interactions at different plant densities changes along stress gradients would be very helpful.

R: We have now added to the figure a Panel B which shows how interactions change along stress gradients and make clear that this is essentially the same information as in panel A but better visualizes the shift in applicability of the SGH along a density

gradient. **(new Fig.1(B), lines 106-108)**

Ln 97, delete “with”

R: Done. **(lines 110)**

Ln 102, has been neglected?

R: We have rephrased this to indicate that facilitative effect and response have rarely been modeled simultaneously. **(lines 124-125)**

Ln 118–122, I think extrapolating to interspecific settings simply based on that intraspecific competition is often stronger than intraspecific competition is more than a stretch.

R: Intraspecific facilitation are usually thought to occur less easily than interspecific facilitation because of the larger niche overlap and stronger intraspecific competition. Therefore, we originally thought that if facilitation could increase with density of conspecific plants (when density is not too high), then the density–dependence should be observed more easily in interspecific cases. This comment is closely related to the major comment above, and we have addressed it there (see above and **lines 139-147, 110-119)**.

Figure 2, why was e^8 the maximum density used in simulations, not the exact same range of density as in the experiment? How were the ranges of density comparable between simulations and experiments? There is a short explanation (Ln 344–345), but not sufficient.

R: We have rewritten the explanation and hope it is clearer now. What we wanted to say is that the model is much more general and encompasses a wider range of densities (we believe that as a model it should be very general), but that the densities used in the experiment fall within that range. Namely, in the control (without stress), the competition intensity at the minimum density in simulations should be lower than that in the experiment, while the intensity at the maximum density is higher than that in the experiment. **(lines 453-456)**

Figure 3, Panel B shows how RII changes with salt stress at some of the lowest and highest densities. But how about intermediate densities? Why was it meaningful to have both densities 2 and 3 in panel B, but not density 7 or 10? If this was to match the densities given in Figure 2, it needs to be explained why the range of plant density differed between the simulations and the experiment

R: The main reason for not showing all densities was simply aesthetic. We found that if all the lines were put together, a reader would not be able to distinguish them anymore. I.e. RII–stress curves of density 7&10 were quite similar to curves of density 15&20. Thus, we only displayed the two lowest and two highest densities. We explained this also in a revised figure caption and made clear that we did not omit the intermediate densities in the model. **(lines 166-170, 187-190)**

Ln 163, Perhaps, the “intermediate” stress here is actually a “high” stress, while the “high” stress here is an “extreme” stress. Note what it really means an intermediate, high, and extreme stress.

R: We have replaced “intermediate” and “high” with “high” and “extreme” here (**lines 197**) and relevant sections in Methods (**lines 458, 506**).

Ln 169–175, Why the authors conducted these analyses and presented these results are not aforementioned, so unclear why it’s necessary to have this paragraph here.

R: We agree that this may be confusing but in fact, reviewer 2 requested as one of two major suggestions that we expand our analyses on that point, i.e. there is a contradiction with the suggestion of reviewer 2. As a compromise, we have retained this part but explained better, why it is there already in the Methods section (**lines 465-468**), so that it does not come as a surprise in the Results.

Ln 194, this is a study on interspecific facilitation.

R: We have distinguished intra– and interspecific studies more clearly. (**lines 257-263**)

Ln 225, this is a very bold statement.

R: We have rewritten the relative section and discussed interspecific cases specifically as suggested (see our response to the first Major Comment and **lines 309-313**).

Reviewer #2:

I find the topic interesting, but I have two major concerns with the approach used in the manuscript.

As is briefly mentioned in lines 103 to 109, the plant–plant interactions are not static through time. However, density is a static variable, and will probably only work as a descriptive variable for annual plants. Consequently, it would have been an advantage to consider plant growth data instead of number of seed as the dependent variable. I would also like to see a thorough discussion of this shortcoming in the data and model development.

R: Thanks for your interest and suggestions. We fully agree the dynamics of facilitation could provide important insights and should be analyzed, too. Plant–plant interactions indeed change as individuals grow, even for a given density level. For instance, both negative and positive interactions, are very weak if plants are very small. In fact, we were somewhat surprised by this comment because our original manuscript did contain a reference to such a complementary analysis, i.e. an extensive analysis of our model using growth (size) as the response variable (supplementary information, former Fig. S5).

A related comment on reviewer 1, who asked us to remove the analysis from the main text, indicated to us that we probably failed in explaining the rationale of the additional analysis and in highlighting that we did this analysis. I.e. we came to the conclusion that reviewer 2 most likely overlooked our analyses due to unfortunate wording from our side.

We analyzed size because we believe that size may also be important for annuals. However, this does not mean that density cannot be viewed as the dependent variable. First of all, density-dependence is important for any plant functional type and growth itself is largely determined by density. Also, interactions–density patterns could become stable as individuals grow and changes of interactions (caused by growth) at each density level may not affect these patterns qualitatively. For example, classical theories of competition are mostly based on density, i.e., density vs. competition. Although competition could be stronger as plants at each density level grow bigger, this does not change the fact that competition intensity increases with density.

In this revised version, we have run additional simulations to redraw the original supplementary figure to show how density–dependence of facilitation changes with time and how facilitation changes with growth (a **new Fig. S6**). To avoid similar confusion of future readers, while keeping the focus on density, we have now better highlighted these additional simulations by adding a short explanation to the Methods (**lines 465-468**) and by better highlighting in the Results (**lines 225-232**). In Discussion, we further discuss more specifically how dynamics of facilitation may help to strengthen our arguments (**lines 274-282**).

“It would have been a clear advantage to fit the data to the model instead of just comparing the results using graphical methods. In a Bayesian framework, the joint posterior distribution of parameters and uncertainties could have provided important insights into the studied mechanisms.”

R: We very much welcomed the suggestion of a more formal analysis that could show the fit between the data and models, and we have used a Bayesian framework for these analyses as suggested.

In Bayesian inference, similar to information theoretic approaches, the ‘fit’ of a model is inferred on a relative scale, i.e. comparison between candidate models. We decided that the most straightforward way of defining and selecting models would be by using our own model (which suggests that both density and stress affect interactions, and that they interact) and compare the fit of the data to models that mimic the assumptions of existing theory. More specifically, these models included either only stress as explanatory variable (which corresponds to the classical facilitation models, i.e. a unidirectional change of RII along a stress gradient- the SGH), or density (corresponding to classical competition models), and both parameters- once without interaction and once with interaction between density and stress (corresponding to our own conceptual model).

We have fitted these models to simulations and experimental data respectively, calculated the posterior and uncertainties of model parameters, investigated the

convergence of Markov chain Monte Carlo (MCMC), compared Bayesian models using Watanabe-Akaike information criterion (WAIC), estimated uncertainty in model selection using model averaging and checked the model predictions **(lines 522-552)**. We found that the full model, i.e. where density, stress, and their interaction are included, fits datasets (for both simulations and experimental) the best. There was a 95% probability that estimated model parameters fall within the calculated credible sets **(lines 206-215, see also the new Table 1 and S1, Fig.4 and S4)**. We therefore believe that this formal test of the model fit indeed greatly helped strengthening our claim and are thus very grateful to reviewer 2 for making this suggestion.

Reviewer #3:

The authors combined an individual based model and a greenhouse experiment to explore how neighbour densities may alter the outcome of plant-plant interactions along an environmental stress gradient.

The authors conclude that 1) the maximum benefit from neighbouring plants (i.e. facilitation) peaks at intermediate plant density; 2) this peak shifts towards higher densities if environmental stress increases. The authors find these results to be robust and universally applicable and propose a conceptual model to generalize these findings.

These claims are strongly overstated. The authors used a very particular experimental set up with individuals of one plant species exposed to increasing levels of salinity and neighbour densities. It is quite obvious that plants can benefit from increasing the number of neighbouring plants when exposed to increasing salinity gradients because salt uptake by neighbours reduces salt concentrations in the soil and therefore ameliorates this abiotic stress. The results from this experiment are very specific to salt and cannot be generalized to other type of abiotic (e.g. drought, nutrient limitation) or biotic stresses (e.g. herbivory).

R: Our response to this comment is two-fold and we have a more conceptual reply regarding the generality of our overall findings and a specific reply regarding only the experimental results.

On a more general level, our statement which was regarded as 'overstated' referred to our overall study, and we politely disagree with the judgement of an overstatement. Our study consists mainly of a simulation model that we have run in many different scenarios, and all the scenarios show the very same pattern. The goal of models in general and of our model, too, is to generate testable predictions for generally applicable principles. In our study, we could show such general patterns, and we elaborate in detail why we believe these patterns to be universally applicable. Usually, modeling studies stop at this point, but we also conducted an experiment to test the model with real plants. The experimental data were strikingly similar to the modeling results, thus further corroborating that the predictions made by the model are not only general but can also be found in real plants.

In response to this more general criticism, we have made this point clearer now, i.e. we explain that our overall conclusion about the robustness and generality of the results applies to the overall study **(lines 295-301)**.

The authors reason that if a benefactor of a given size alleviates stress to a certain level, then more plants are needed to generate the same conditions in a more severe environment. This explanation is quite specific to stresses such as salt and perhaps some pollutants but not for other abiotic stresses. For example, if the authors would have exposed plants of the same size to increasing neighbour densities along a soil water gradient they would have found increasing competitive effects as soil water decreases.

R: While we believe that our answer above may remedy the main concerns, we also felt we should respond to the very specific comments on the experimental part of our study. Here, reviewer 3 had two major concerns: 1) the study only tested salt stress, which is specific; 2) more neighbors will lead to stronger competition instead of facilitation with increasing water deficiency, namely, the density–dependence of facilitation cannot be observed in other stressful conditions.

Our answer to this first concern is again two-fold: Firstly, as said above, we performed a simulation study and found a generally applicable pattern, i.e. the model is not at all specific for a salinity gradient. At the same time, we believe that our model could, and should, inspire many more experimental studies with more species and stress factors, and we spell this out more prominently. Obviously, our study is a proof of principle, and cannot conduct several experiments with many species and stress factors at the same time.

We would also like to point out that this reviewer’s assumption that plants will compete more when (drought) stress is higher is in direct contradiction to the stress gradient hypothesis, i.e. the most pertinent hypothesis in facilitation theory. This hypothesis has been phrased for any type of stress factors, including drought, substrate stability, herbivory or nutrient limitation. We also believe that when neighbors increase water or nutrient levels or protect from herbivory, then this effect will be larger when there are more neighbors, because more neighbors shade more of the substrate (alleviation of drought stress or salinity stress), release more nutrients (alleviation of low nutrient stress), stabilize mobile substrate more or deter herbivores more. I.e. we do not agree that density-dependent facilitation should be exclusive to salt stress, and we do not agree that the SGH applies only to salinity.

Fortunately, despite the fact that density is underexplored in facilitation research, we found some evidence from water deficiency or other stress factors that show that our model predictions are not limited to salt or soil contaminants. For example, Goldenheim et.al found that under stronger evaporative stress, growth and seed production of *Suaeda linearis* (forb) increased with the density of conspecific neighbors, indicating stronger facilitation at high densities. However, under milder conditions, these plants exhibited negative density–dependence, i.e., stronger competition at higher densities. They also reported that dense stands could significantly reduce desiccation and thermal stress (Goldenheim et.al, 2008). Similarly,

Fajardo and McIntire found that *Nothofagus pumilio* (tree, seedling) exhibited stronger facilitation (survival) at higher densities because neighbors may reduce water losses caused by wind and radiation (Fajardo and McIntire, 2011). Accordingly, we had mentioned in the original manuscript other stress factors that should be very similar to salinity. We have now expanded the number of previous empirical studies which we discuss and that have used different stress factors. We have also expanded this paragraph in the discussion to acknowledge a large range of harsh conditions and facilitation mechanisms all of which should be positively affected by density (until competition kicks in). We hope that by greatly expanding the discussion about other factors, the concerns of reviewer 3 are remedied **(lines 295-313)**.

REVIEWERS' COMMENTS:

Reviewer #1 (Remarks to the Author):

In my view, the manuscript has been greatly improved. Thanks to the authors for these efforts. I am happy with this version of the manuscript except for the following two places.

1) To be consistent, there need to be four levels of stress in Fig. 1B

2) in response to one of my previous comments, the authors explained that "the densities used in the experiment fall within that range". But this is not really the case since the densities in the simulation ranged from 7 to 2980 while those in the experiment from 1 to 20. Why was 2980 considered to be a reasonably high density in the simulation while a way smaller number - 20 - to be a reasonably high density in the experiment? These need further clarification.

Reviewer #2 (Remarks to the Author):

[Editor's note: the referee did not leave remarks to the authors, but left remarks to the editors indicating satisfaction with the revisions.]

Point-by-point response to reviews (reviews in *italics*, response indicated by 'R'):

Reviewer #1 (Remarks to the Author):

In my view, the manuscript has been greatly improved. Thanks to the authors for these efforts. I am happy with this version of the manuscript except for the following two places.

1) To be consistent, there need to be four levels of stress in Fig. 1B

R: We have added two levels of stress and redrawn the figure. (**see new Fig. 1B**)

2) in response to one of my previous comments, the authors explained that "the densities used in the experiment fall within that range". But this is not really the case since the densities in the simulation ranged from 7 to 2980 while those in the experiment from 1 to 20. Why was 2980 considered to be a reasonably high density in the simulation while a way smaller number - 20 - to be a reasonably high density in the experiment? These need further clarification.

R: Indeed, 20 is a way smaller number when compared with 2980. However, we should consider that the scale of the simulation is more likely to be '1 m²' while the pots used in the experiment are samll (10×10cm). In fact, 20 individuals per pot is equivalent to 2000 ind. per m² and this is very dense. We have added a short explanation to the Methods to avoid such confusions. (**Line 468-469**)